# A Vegan Athlete’s Heart—Is It Different? Morphology and Function in Echocardiography

**DOI:** 10.3390/diagnostics10070477

**Published:** 2020-07-14

**Authors:** Wojciech Król, Szymon Price, Daniel Śliż, Damian Parol, Marcin Konopka, Artur Mamcarz, Marcin Wełnicki, Wojciech Braksator

**Affiliations:** 1Department of Sports Cardiology and Noninvasive Cardiovascular Imagining, Medical University of Warsaw, 02-091 Warszawa, Poland; wojciech.krol@wum.edu.pl (W.K.); szymonprice@gmail.com (S.P.); marcin.konopka@wum.edu.pl (M.K.); wojciech.braksator@wum.edu.pl (W.B.); 23rd Department of Internal Medicine and Cardiology, Medical University of Warsaw, 02-091 Warszawa, Poland; damian.parol@wum.edu.pl (D.P.); artur.mamcarz@wum.edu.pl (A.M.); marcin.welnicki@wum.edu.pl (M.W.)

**Keywords:** echocardiography, vegan, athletes’ hearts, runners, diet

## Abstract

Plant-based diets are a growing trend, including among athletes. This study compares the differences in physical performance and heart morphology and function between vegan and omnivorous amateur runners. A study group and a matched control group were recruited comprising *N* = 30 participants each. Eight members of the study group were excluded, leaving *N* = 22 participants. Members of both groups were of similar age and trained with similar frequency and intensity. Vegans displayed a higher VO2max (54.08 vs. 50.10 mL/kg/min, *p* < 0.05), which correlated positively with carbohydrate intake (ρ = 0.52) and negatively with MUFA (monounsaturated fatty acids) intake (ρ = −0.43). The vegans presented a more eccentric form of remodelling with greater left ventricular end diastolic diameter (LVEDd, 2.93 vs. 2.81 cm/m^2^, *p* = 0.04) and a lower relative wall thickness (RWT, 0.39 vs. 0.42, *p* = 0.04) and left ventricular mass (LVM, 190 vs. 210 g, *p* = 0.01). The left ventricular mass index (LVMI) was similar (108 vs. 115 g/m^2^, *p* = NS). Longitudinal strain was higher in the vegan group (−20.5 vs. −19.6%, *p* = 0.04), suggesting better systolic function. Higher E-wave velocities (87 vs. 78 cm/s, *p* = 0.001) and E/e′ ratios (6.32 vs. 5.6, *p* = 0.03) may suggest better diastolic function in the vegan group. The results demonstrate that following a plant-based diet does not impair amateur athletes’ performance and influences both morphological and functional heart remodelling. The lower RWT and better LV systolic and diastolic function are most likely positive echocardiographic findings.

## 1. Introduction

The vegan diet is one of the fastest-growing trends in nutrition [1]. Between 2014 and 2018, the number of followers of the vegan diet has increased by 600% in the US [2]. This causes both easier access to high-quality vegan products, as well as more research on plant-based diets and better access to knowledge about supplementation and the proper balancing of such diets. This trend can also be observed in the health and fitness industries. Many athletes decide to change their diet and many recent studies focused on the impact this has on their performance.

Regular amateur or professional endurance training induces many morphological and functional adaptations in the cardiovascular system. They include increased dimensions of the heart’s chambers and increased wall thickness and muscle mass. Such changes may mimic those observed in pathological conditions and sometimes require monitoring.

In echocardiographic examinations, endurance-trained athletes demonstrate an increased relative wall thickness (RWT), left ventricular mass (LVM), left ventricular end-diastolic internal diameter (LVIDd) and left atrial volume index (LAVI) [3]. The ejection fraction, however, remained similar to untrained control groups. Newer methods using speckle tracking methods to establish global longitudinal strain (GLS) may present more accurate data on the heart’s functioning. Recent research shows that GLS may be related to different levels of exercise among athletes and may be different in athletes compared to in healthy controls [4,5,6].

So far, very little information is available on how a vegan diet affects athletes’ hearts. The only research on the impact of a plant-based diet on the cardiovascular system addressed heart failure and other diseases [7,8]. The findings of these studies may suggest that it could potentially be beneficial for athletes. To the best of our knowledge, no study compared the echocardiographic parameters of vegan and non-vegan athletes. The aim of this study was to assess the differences in the athletes’ heart morphology and function and the correlation of these with dietary habits.

## 2. Method

### 2.1. Subjects

A study and control group were recruited. The participants were recruited from organized amateur running events, such as the Warsaw Marathon, and by online invitation published on social media. Inclusion criteria for the study group (vegan—V) were: having completed at least one organized running event with a distance of at least 10 km, declaring a vegan diet and regular training at least three times a week. The same criteria (excluding the vegan diet) applied to the control group (C), which was age-matched to the study group. The declared vegan diet was then verified by a dietetic survey and nutrition diary. Exclusion criteria for both groups were: age below 18 years, known cardiovascular or respiratory disease and taking any medication on a regular basis. Initially, both groups comprised 30 athletes each. Ultimately, six participants from the study group were excluded due to not following a vegan diet after the dietician’s verification and two were excluded due to a suboptimal acoustic window in echocardiography, which did not allow us to perform the required measurements accurately. As a result, the final study group consisted of 22 athletes, and the control group consisted of 30. This study uses the study population from previously published extensive research on plant-based diets in endurance athletes [9].

The study was approved by the Bioethics Committee of the Medical University of Warsaw: No. KB/214/2014 of 4 November 2014 with Annex No. KB/34/A/2015 of 6 May 2015. Written consent was acquired from all participants.

### 2.2. Tests

The tests carried out among all the participants included: anthropometric measurements (weight, height, skinfold measurement, body composition evaluation using bioimpedance analysis), complex dietetic evaluation based on a 4-day-long nutrition diary, spiroergometric testing on a treadmill and resting echocardiographic tests following current guidelines [10], including global longitudinal strain (GLS) evaluation using speckle tracking methods. In order to accommodate expected differences in height and weight among the participants, the parameters were indexed to the body surface area (BSA) calculated using Mosteller’s formula [11].

The echocardiographic measurements were carried out on a GE Vivid 6 echocardiograph using a 4S sector transducer. All the measurements were performed, recorded and evaluated offline using EchoPac (ver. 112 GE, USA) software by one experienced echocardiographer (WK). The chamber size measurements were performed as recommended [10]. The echocardiographer was blinded to the patients’ classifications to the V or C group. Segmental strains were calculated using utomated Function imaging (AFI, GE, USA) to obtain GLS measurements. The software automatically divided the left ventricle wall into 17 segments based on manually selected points, two basal and one apical. After the careful assessment of the adequacy of tissue tracing and any necessary adjustments, the software presented the strain curves of all 17 segments and reported peek systolic values. GLS was defined as the average of all examined segments. Following current recommendations, the results concerning LV strain were analysed in absolute values (higher representing better systolic function) [6].

### 2.3. Statistical Analysis

For the statistical analysis, the commercially available software STATISTICA ver. 13.3 (StatSoft, Tulsa, OK, USA) was used. Continuous variables are presented as mean ± standard deviation (SD), and categorical variables as percentages. The normal distribution of all continuous variables was examined using the Shapiro–Wilk test. The unpaired *t*-test and the Mann–Whitney U test were used according to data distribution to assess differences between groups in continuous variables and the chi-square test was used for categorical variables. The correlations between continuous variables with normal distributions were assessed using the Pearson correlation coefficient R. The Spearman rank correlation coefficient was used for categorical variables or those with non-normal distributions. Correlations were analysed for all participants together (*N* = 52).

## 3. Results

### 3.1. General Characteristics

The general characteristics of the participants are demonstrated in Table 1. The observed insignificant difference in age is a result of excluding eight members of the V group.

Athletes from the V group weighed significantly less, while being similarly tall. This resulted in lower BMI and BSA. The number of hours spent on training weekly was similar, as was the number of kilometres covered weekly.

### 3.2. Diet

The daily energy intake for both groups was similar (2647 ± 618 vs. 2408 ± 557 kcal/d, *p* = 0.051). The percentage of energy from protein was significantly lower in the V group (11.8 ± 1.9 vs. 18.1 ± 3.32%, *p* < 0.0001), as was the percentage of energy from fat (25.6 ± 9.8 vs. 31.7 ± 6.6%, *p* = 0.0006). The percentage of energy from carbohydrates was higher in the V group (61.7 ± 11.1 vs. 49.0 ± 7.9%, *p* < 0.0001). Vegans had a lower absolute (g) intake of saturated fatty acids (SFAs, 13.2 ± 7.2 vs. 30.9 ± 11.4 g, *p* < 0.0001), and a higher intake of polyunsaturated fatty acids (PUFAs, 25.7 ± 11.9 vs. 13.8 ± 5.9 g, *p* < 0.0001). The absolute intake of monounsaturated fatty acids (MUFAs) was similar (30.42 ± 15.6 vs. 35.1 ± 12.5 g, *p* = 0.11), however, the percentage of the daily energy intake from MUFAs was significantly higher in the V group (13 ± 3 vs. 11 ± 5%; *p* = 0.004).

### 3.3. Performance

The peak power output (measured in watts) reached in the treadmill test was similar in both groups. Performance parameters evaluated in the spiroergometry test are presented in Table 2. The absolute exercise capacity, measured as maximal oxygen consumption (VO2max, L/min), was similar in both groups, however the VO2max per kilogram of body mass (mL/min/kg) was higher in the V group. VO2max (mL/min/kg) correlated significantly with the total and per kilogram intake of carbohydrates (ρ = 0.43, ρ = 0.52, respectively) and correlated negatively with the intake of MUFAs (ρ = −0.43).

### 3.4. Echocardiographic Findings

The morphological findings are demonstrated in Table 3. Vegans presented a more eccentric form of left ventricular remodeling, with greater left ventricular end diastolic diameter (LVEDd) and thinner LV walls (both the intraventricular septum in diastole (IVSd) and the posterior wall in diastole (PWd)). This resulted in a lower relative wall thickness. Both LVM and LVMI correlated negatively with PUFA intake (ρ = −0.38, ρ = −0.37, respectively). No significant differences were noticed in left atrium (LA) and right atrium (RA) sizes. LA enlargement (>34 mL/m^2^) was present in nearly 70% of athletes. Functional systolic and diastolic findings are shown in Table 4. Vegans displayed a higher GLS and higher E-wave velocities of mitral inflow, resulting in a higher E/e′. The GLS correlated positively with SFA and MUFA intakes (ρ = 0.34, *r* = 0.31, respectively). The E velocity correlated positively with the intake of plant proteins and carbohydrates (ρ = 0.32, ρ = 0.44, respectively) and correlated negatively with the intake of SFAs per kilogram of body mass (ρ = −0.036).

## 4. Discussion

This study presents detailed echocardiographic examination results in a group of vegan athletes and compares them with a well-matched control of athletes on a mixed diet. An additional strength of the study is the wide array of parameters measured in both groups, including detailed dietary data, which allow a more in-depth interpretation of the results.

Overall, the hearts in the V group presented remodelling defined classically as more typical for endurance training—more eccentric, with thinner walls and a larger diameter of the left ventricle [12]. Importantly, despite the ventricle being larger, the LVM was lower in the V group. It has been reported that a higher left ventricular mass in runners is associated with a higher coronary artery calcium (CAC) score, which in the general population is associated with the occurrence of major cardiovascular events. Such a correlation with an increased cardiovascular risk was not confirmed in athletes, although the phenomenon is not yet well described [13,14,15,16]. Concentric remodelling and LV hypertrophy are both associated with a higher risk of all-cause mortality [17]. Therefore, the remodelling present in the V group may be more physiologic.

Differences in the diastolic function were also found, measured with tissue Doppler methods (MV E velocity and E/e′). In pathological conditions, such as in patients with hypertension or reduced left ventricular ejection fraction (LVEF), the increased E/e′ ratio suggests an elevated filling pressure in the left ventricle [18]. These measurements, however, are not accurate in predicting the filling pressure in normal, healthy subjects [18]. Recent studies confirmed that the athlete’s heart is a physiological condition and is not associated with fibrosis or increased filling pressure [19]. In pathologic conditions, decreased E/e′ is predominantly caused by lower e′ velocities due to the fibrosis of myocardial tissue. Increased E-wave velocity is later present in advanced diastolic disfunction (restriction). On the contrary, in athletes and in our examined group, high E-wave velocity was concomitant with high e′ velocity. In this setting, high E-wave velocity is an indicator of dynamic, efficient inflow during early diastole. It has been demonstrated that better endothelial function and oxidative stress parameters in athletes play an important role in physiological LV remodelling associated with better subendocardial function due to an optimized ventriculo-arterial coupling [19]. The higher E/e′ ratio and MV E velocity may therefore be considered a consequence of a more dynamic diastole in a physiological LV. A vegan diet results in lower oxidative stress and may improve endothelial function [8,20,21]. This might be responsible for the better diastolic function in the V group. The correlations of the MV E velocity with plant protein and carbohydrate intakes could suggest a potential influence of these dietary ingredients on a better diastolic function. On the other hand, the higher intake of SFAs may be responsible for the slightly worse diastolic function in non-vegans, based on the negative correlation. A recent pilot study examined the effects of increasing the dietary intake of unsaturated fatty acids in nine individuals with heart failure with preserved ejection fraction (HFpEF) and obesity [22]. The only intervention was to increase the intake of UFA-rich foods (canola, olive oil, tree nuts, peanuts), for 12 weeks without recommendations on energy intake. Aerobic capacity was tested at baseline and at 12 weeks using a treadmill metabolic cart. The authors observed a significant improvement in exercise time and O_2_ pulse with a trend toward a significant increase in peak VO_2_ (*p* = 0.069). Changes in peak VO_2_ tended to associate with changes in plasma UFAs (R = +0.71; *p* = 0.071). The statistically non-significant result may be due to the small sample size of the pilot study. These results could suggest that the better performance of the vegan group might be due to the higher UFA consumption. Carbone et al. did not perform echocardiography in this pilot study, but their results, together with the results of the present study, may suggest that the improvement in cardiorespiratory fitness is at least partly due to improved cardiac function.

The echocardiographic evaluation of athletes’ hearts remains a difficult issue due to the potential overlap of physiological adaptation and pathological conditions [23]. While the LVEF usually remains normal, the echocardiographic diameters of heart chambers may still meet the diagnostic criteria for many serious diseases, such as arrhythmogenic cardiomyopathy and dilated cardiomyopathy [23]. This does not only affect professional athletes, as even moderate training leads to significant changes in the heart and increases the risk of meeting the diagnostic criteria of LV hypertrophy or dilation and RV dilation in cardiac magnetic resonance (CMR) imaging [24]. New techniques, such as GLS evaluation, could give more insight into the function of the athlete’s heart [25]. Unfortunately, the results of studies on this parameter are highly controversial. Some evidence suggests that professional athletes have lower GLS compared to controls and recreational athletes [4,6], but another trial yielded opposite results [5]. A recent meta-analysis did not reveal differences in GLS between athletes and controls, but the results were limited by study heterogeneity [26]. Despite these differences, authors generally agree that lower GLS values can be an early marker of LV systolic dysfunction [27]. In this study, the GLS in both groups fell close to the proposed normal value of 19.7% (95% CI, 20.4% to 18.9%) [28]. The strain in the V group was significantly higher, in conjunction with better exercise capability (expressed in VO2max). The correlation between GLS and SFA and MUFA intake could suggest a cause for this difference. This difference could also potentially be related to lower cholesterol levels and better glucose metabolism parameters, which are beneficial for the heart muscle. Such a relationship was demonstrated in type 1 and 2 diabetes patients—GLS was significantly higher with increasing levels of cholesterol remnants and triglycerides [29].

The large proportion of athletes with an enlarged LA in our study is similar to recent studies and is most likely not an indication of pathology, but a physiological adaptation. Even in athletes with advanced atrial remodelling, the left atrial function remains normal, similarly to the left ventricle. Moreover, it has been shown that left atrial volume correlated with exercise capacity in professional athletes [30,31].

The higher VO2max reached by the vegan athletes may imply that they are better trained than the control, thus impacting the echocardiographic findings. However, the weekly training frequency and running distances were similar in both groups, suggesting that other factors may be responsible for the higher VO2max. The moderate correlation of VO2max with the carbohydrate intake and inverse correlation with SFA intake suggests that these dietetic factors may be partly responsible for the difference. It has been found that higher carbohydrate consumption is associated with better performance in an intermittent exercise test [32]. The carbohydrate consumption in the V group in our study (62% of energy) was significantly higher than in the C group, similar to the carbohydrate-enriched diet in the study conducted by Bangsbo et al. (65% of energy) [32]. Our result is also consistent with a recent study which demonstrated that vegetarian athletes had a higher VO2max than omnivorous athletes [33]. Another factor to consider is the similar maximum power output achieved in both groups, despite the higher VO2max in the V group.

## 5. Conclusions

A vegan diet does not result in impaired performance in amateur runners.Following a plant-based diet may influence both morphological and functional heart remodelling.The vegan diet may be associated with certain, most likely positive, characteristics in echocardiography (lower RWT, better LV systolic and diastolic function).

## Figures and Tables

**Table 1 diagnostics-10-00477-t001:** General characteristics.

Group	V	C
Age (years)	32 ± 5	30 ± 5
Height (cm)	178.5 ± 7	180.5 ± 7
Weight (kg)	68.6 ± 7 *	75.1 ± 6 *
BSA (m^2^)	1.75 ± 0.1 *	1.83 ± 0.1 *
BMI (kg/m^2^)	21.6 ± 2.1 *	23 ± 1.3 *
Weekly practice time (h)	5.5 ± 4	4.9 ± 2
Weekly distance (km)	48.7 ± 3	48.5 ± 21
Training experience (years)	4.9 ± 4	3.9 ± 3

* *p* < 0.05; BSA—body surface area; BMI—body mass index.

**Table 2 diagnostics-10-00477-t002:** Performance parameters.

Group	V	C
VO2max (L/min)	3.70 ± 0.5	3.75 ± 0.6
VO2 AT (L/min)	2.24 ± 0.7	2.29 ± 0.7
VO2max (mL/kg/min)	54.0 ± 7.0 *	50.1 ± 7.2 *
Power max (W)	309 ± 36	324.2 ± 40.3
Power AT (W)	197 ± 48	216 ± 49
HR max	167 ± 29	168 ± 27
HR AT	144 ± 24	142 ± 19
SBP max (mmHg)	158 ± 30	157 ± 26
DBP max (mmHg)	86.8 ± 10	83 ± 21

* *p* < 0.05; VO2—rate of oxygen consumption; max—measured at peak performance; AT—measured at anaerobic threshold; HR—heart rate; SBP—systolic blood pressure; DBP—diastolic blood pressure.

**Table 3 diagnostics-10-00477-t003:** Echocardiographic morphology parameters (with and without BSA indexation).

Group	V	C	*p*-Value
LVIDd (cm)	5.12 ± 0.2	5.11 ± 0.2	NS
LVIDd/BSA (cm/m^2^)	2.93 ± 0.3	2.81 ± 0.2	0.04
IVSd (cm)	1.00 ± 0.10	1.08 ± 0.1	0.01
IVSd/BSA (cm/m^2^)	0.58 ± 0.1	0.59 ± 0.1	NS
RVOT	2.92 ± 0.2	2.89 ± 0.3	NS
RVOT/BSA (cm/m^2^)	2.3 ± 0.2	2.1 ± 0.2	0.003
LVM (g)	190 ± 34	210 ± 31	0.01
LVMI (g/m^2^)	108 ± 17	115 ± 14	NS
RWT	0.39 ± 0.07	0.42 ± 0.06	0.03
LAV (mL)	66.5 ± 19	74.6 ± 16	NS
LAV/BSA (mL/m^2^)	38 ± 10	40.3 ± 10	NS
RAA/BSA (cm^2^/m^2^)	11.9 ± 2.7	11.1 ± 2.2	NS

LVIDd—left ventricular internal dimension at end-diastole; BSA—body surface area indexation; IVSd—interventricular septum thickness at end-diastole; RVOT—right ventricular outflow tract; LVM—left ventricular mass; LVMI—left ventricular mass index; RWT—relative wall thickness; LAV—left atrial volume; RAA—right atrial area.

**Table 4 diagnostics-10-00477-t004:** Systolic and diastolic echocardiographic parameters.

Group	V	C	*p*-Value
E (m/s)	0.87 ± 0.1	0.79 ± 0.1	0.02
A (m/s)	0.44 ± 1.1	0.44 ± 1.4	NS
e′ ivs (cm/s)	14.2 ± 2.7	14.3 ± 2.5	NS
e′ lat (cm/s)	19.6 ± 3.5	19.1 ± 3.1	NS
E/e′	6.3 ± 1.3	5.60 ± 1	0.03
TVs′	16.1 ± 2.8	15.4 ± 2.8	NS
Peak GLS (%)	20.5 ± 2.2	19.6 ± 1.5	0.04

E—peak velocity of early diastolic transmitral flow; A—peak velocity of late transmitral flow; e′—peak velocity of early diastolic mitral annular motion as determined by pulsed wave Doppler; TVs’—peak velocity of systolic tricuspid annular motion as determined by pulsed wave Doppler; GLS—global longitudinal strain.

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
