# Peer review of "A Vegan Athlete’s Heart—Is It Different? Morphology and Function in Echocardiography"

_diagnostics, 2020, doi:10.3390/diagnostics10070477_

Round 1
Reviewer 1 Report
This is a timely and interesting topic in the general population as well as in athletes. It is refreshing to see a manuscript focused on the health of the heart rather than simply avoidance of disease. This manuscript is well written and easy to understand. The methods appear to be appropriate, though I am not able to assess the echocardiographic methods.
3.2 Diet
There is a character error in the first sentence.
I would love to have seen a break down of protein intake to assess intake of essential amino acids, which is the typical concern with the vegan diet. This is not necesssary but would be extremely interesting to the nutritionists. It may strengthen your manuscript.
Discussion
Line 164: This sentence is unclear to me, and this seems to be an important point.
Line 173 A plant-based diet and a vegan diet are not necessarily the same thing. Switching terminiologies is confusing the matter.
Line 185 "Statistical significance might not have" -- Very confusing wording. Makes it sound like you are not sure if it was significant rather than if that was due to power.
Line 187 VEG has not been defined & "et al." has a period at the end
Line 211-12 How so?
Conclusions
2. You need to use the term vegan here, as this is what you assessed, not a plant-based diet, which includes vegetarians and omnivores with low animal product intake. The phrasing is also a bit strong. Can you definitively say that a viegan diet influence both with this study? Using the word may is most likley more appropriate.
3. This needs to be read aloud and rephrased. Similar strength of conclusions issues to the previous conclusions, consider using the word may or similar phraseology.
Author Response
This is a timely and interesting topic in the general population as well as in athletes. It is refreshing to see a manuscript focused on the health of the heart rather than simply avoidance of disease. This manuscript is well written and easy to understand. The methods appear to be appropriate, though I am not able to assess the echocardiographic methods.
We have made the echocardiographics description more precise. All the measurements were carried out according to guidelines, which are cited in the references.
3.2 Diet
There is a character error in the first sentence. –
Corrected.
I would love to have seen a break down of protein intake to assess intake of essential amino acids, which is the typical concern with the vegan diet. This is not necesssary but would be extremely interesting to the nutritionists. It may strengthen your manuscript.
An amino acid breakdown has been done for the participants, but we did not include it in the results because we considered that the amount of data would be too large and confounding, and would negatively affect the clarity of the article. The article focuses on the echocardiographic imaging, and for many readers this dietetic information could be too detailed. If you wish, we can upload a table with the amino acids as supplementary material.
Discussion
Line 164: This sentence is unclear to me, and this seems to be an important point.
We have discussed the significance of E/e’ differences in more detail, we hope this is now clear.
Line 173 A plant-based diet and a vegan diet are not necessarily the same thing. Switching terminiologies is confusing the matter. - Corrected
Line 185 "Statistical significance might not have" -- Very confusing wording. Makes it sound like you are not sure if it was significant rather than if that was due to power. – Wording changed
Line 187 VEG has not been defined & "et al." has a period at the end - corrected
Line 211-12 How so?
Even in athletes with advanced atrial remodeling, the left atrial function remains normal, similarly, to the left ventricle. What is moreMoreover, it has been shown that left atrial volume correlated with exercise capacity in professional athletes [30][31].
Conclusions
2. You need to use the term vegan here, as this is what you assessed, not a plant-based diet, which includes vegetarians and omnivores with low animal product intake. The phrasing is also a bit strong. Can you definitively say that a viegan diet influence both with this study? Using the word may is most likley more appropriate. – Wording changed
- This needs to be read aloud and rephrased. Similar strength of conclusions issues to the previous conclusions, consider using the word may or similar phraseology. – The sentence has been rephrased for more clarity and with the word ‘may’.
Reviewer 2 Report
Presented study had a small number of patients and provided data was obtained from single-centre. Thus, data obtained from this study have several limitations. The manuscript is well written, however seems to be not merit to be publish in this journal.
Author Response
Presented study had a small number of patients and provided data was obtained from single-centre. Thus, data obtained from this study have several limitations. The manuscript is well written, however seems to be not merit to be publish in this journal.
Response:
The study group was small, but the results we obtained were statistically significant. It would be difficult to study a larger group for several reasons:
1 difficulty in recruiting suitable volunteers in large numbers, as the group we selected was very specific
2 High costs and labour intensity of the trial – we took great care to ensure our results are accurate. The tests were performed on a high-end echocardiograph by experienced specialists in the field of sports cardiology and dietetic information was gathered and analysed by professional dieticians.
3 The limited research available on the topic and lack of substantial evidence supporting our hypotheses that the heart morphology and function would be different in vegans made it difficult to obtain funding for large, multi-centre studies.
Despite these difficulties we did, as mentioned above, achieve significant and, we believe, important results. We hope that our evidence will also be of use to design larger, perhaps prospective multi-centre studies, for we agree that such studies would provide stronger evidence.
Reviewer 3 Report
The impact of vegan diet on the exercise performance and cardiovascular system is receiving great concern. The authors investigated the association between the vegan diet and morphology and function of athletes’ hearts. There are several concerns.
- Given the inclusion criteria, the authors’ findings are adopted only for the runner athletes?
- Is it possible to a little bit more detail on how and where the study group was collected?
- Is it sufficient to match only age?
- Informed consent was written from participants?
- Numerically, there seems to be differences in several parameters in Table 1. These differences are really not significant?
- Correlation analyses were performed among all participants or only among the vegan group?
- One of the limitations of this study might be a lack of any outcome data.
- Is it possible to explain the mechanism of why the vegan diet has considerable impacts on cardiac function?
Author Response
The impact of vegan diet on the exercise performance and cardiovascular system is receiving great concern. The authors investigated the association between the vegan diet and morphology and function of athletes’ hearts. There are several concerns.
1. Given the inclusion criteria, the authors’ findings are adopted only for the runner athletes?
The recruitment criteria, as mentioned in the text, required that the participants regularly trained running. However, these were amateurs and as such many of them did not focus solely on running. The results are most accurate for amateur vegan runners, but could probably be extrapolated to most amateur endurance athletes.
2. Is it possible to a little bit more detail on how and where the study group was collected?
The participants were recruited from organized amateur running events, such as the Warsaw Marathon, and online.
3. Is it sufficient to match only age?
The group was also adjusted for gender (only males were recruited). Apart from matching for age and gender, most calculations were also adjusted for weight.
4. Informed consent was written from participants?
Yes.
5. Numerically, there seems to be differences in several parameters in Table 1. These differences are really not significant?
The differences which are significant have been labelled with an asterisk. The other results are non-significant (we double-checked the calculations after this review), with p>0.05.
6. Correlation analyses were performed among all participants or only among the vegan group?
Among all participants – text modified.
7. One of the limitations of this study might be a lack of any outcome data.
This is true, unfortunately we were not able to provide outcome data in this initial trial. We hope that our data will provide the basis for planning larger prospective studies.
8. Is it possible to explain the mechanism of why the vegan diet has considerable impacts on cardiac function?
We attempted to explain the differences with different intake of saturated and unsaturated fatty acids, as well as protein. So far, data on this topic is very limited and more research in the field would be required to further explain the differences, including prospective studies.
Round 2
Reviewer 2 Report
This article is not merit for publication in this journal.